# Three-Dimensional Bioprinting of Naturally Derived Hydrogels for the Production of Biomimetic Living Tissues: Benefits and Challenges

**DOI:** 10.3390/biomedicines11061742

**Published:** 2023-06-17

**Authors:** Elena Merotto, Piero G. Pavan, Martina Piccoli

**Affiliations:** 1Tissue Engineering Lab, Istituto di Ricerca Pediatrica Città della Speranza, Corso Statu Uniti 4, 35127 Padova, Italy; elena.merotto@studenti.unipd.it (E.M.); piero.pavan@unipd.it (P.G.P.); 2Department of Industrial Engineering, University of Padova, Via Gradenigo 6a, 35129 Padova, Italy

**Keywords:** tissue engineering, 3D bioprinting, naturally derived hydrogel, hydrogel crosslinking

## Abstract

Three-dimensional bioprinting is the process of manipulating cell-laden bioinks to fabricate living structures. Three-dimensional bioprinting techniques have brought considerable innovation in biomedicine, especially in the field of tissue engineering, allowing the production of 3D organ and tissue models for in vivo transplantation purposes or for in-depth and precise in vitro analyses. Naturally derived hydrogels, especially those obtained from the decellularization of biological tissues, are promising bioinks for 3D printing purposes, as they present the best biocompatibility characteristics. Despite this, many natural hydrogels do not possess the necessary mechanical properties to allow a simple and immediate application in the 3D printing process. In this review, we focus on the bioactive and mechanical characteristics that natural hydrogels may possess to allow efficient production of organs and tissues for biomedical applications, emphasizing the reinforcement techniques to improve their biomechanical properties.

## 1. Introduction

In 1993 Langer and Vacanti defined tissue engineering as “an interdisciplinary field that applies the principles of engineering and the life sciences toward the development of biological substitutes that restore, maintain, or improve tissue function” [1]. The original aim of tissue engineering was, indeed, the in vitro generation of organs or tissues to be used as in vivo substitutes to shorten the lists of patients waiting for a transplant. Since then, many attempts have been made to reconstruct several organs in the laboratory using different techniques and materials [2,3,4]. Some of these have even led to important clinical successes [5,6], but the paradigm of tissue engineering changed completely when the 3D printing technique was introduced to the biomedical field. 

By definition, 3D bioprinting is the use of computer-aided transfer processes to pattern and assembling living and non-living materials with precise organization [7]. Due to this ability and the consequent possibility of achieving the distribution of cells in a different time and space, 3D bioprinting has become the ideal technique to generate 3D living structures in vitro. The benefits of this organ and tissue manufacturing include: (i) the generation of 3D structures with defined shape, size, and geometry [8]; (ii) the proper distribution and positioning of biomaterials, signaling factors, and different types of cells in high densities; (iii) spatial depth and better cell–cell communication for improved physiology [9]. All of these aspects not only make 3D bioprinting the means that, in the future, will allow the construction of patient-specific organs useful for transplantation purposes, but it is an in vitro approach that exceeds standard 2D culture techniques [10] and can also eliminate the usual adoption for animal tests, also avoiding the limited accuracy in predicting human toxicological and pathophysiological responses [11]. Until now it has not been possible to print fully functional organs due to their complexity, reflected in the biological variety of cellular components and functions, and in the unrevealed mechanism of growth and development. Nevertheless, this technology has already opened wide spaces for in vitro investigation, with the manufacture of 3D living structures to study the relationship among cells, between cells and the environment, and their response to compounds such as drugs and therapeutic molecules [12]. Traditional methods of 2D cell culture or animal experiments applied for drug screening have many weaknesses. The human environment is far more complicated than the 2D cell condition, where cell behaviors might differ to that observed in vivo. Moreover, the huge difference between animals and humans makes the need for more accurate in vitro models that 3D bioprinting allows to deal with [12].

Although the term 3D bioprinting is often intended for a wide range of biomedical applications, including additive manufacturing of 3D scaffolds that can instruct or induce cells to develop precise features, it specifically refers to the use of computer-aided transfer processes for the production of bio-engineered structures [13]. In this view, the highest and most complex meaning of the term is the manipulation of biologic inks and living cells to build 3D biomimetic tissues. In this sense, 3D bioprinting is generally applied with similar techniques in all laboratories. Extrusion-based bioprinting is the most widely used approach to 3D bioprinting due to its versatility and affordability. This approach can be used to print biomaterials with a wide range of viscosities and different concentrations of cells [14]. Today, the biomaterials mostly used in 3D bioprinting applications are hydrogels: polymeric materials with a hydrophilic structure capable of holding large amounts of water. Natural or synthetic hydrogels can be used as bioinks, i.e., they can be loaded with cells, and extruded using a 3D printer nozzle, due to their suitable mechanical properties. 

In this review, we focus on the benefits and challenges of using naturally derived hydrogels as bioinks for 3D printing approaches and tissue-like construct production, specifically highlighting their biological and mechanical characteristics. Given the wide range of available biological materials and the different properties of each individual biomaterial, we summarize the most popular physical, chemical and natural crosslinking options to modify the properties of naturally derived hydrogels and better suit the 3D bioprinting process. 

## 2. 3D Bioprinting Technology

3D bioprinting technologies for organ manufacturing have improved some traditional medical approaches, especially for fast, precise, and customized biomedical applications [15]. A major goal of bioprinting is the ability to customize the size and shape of the printed constructs to best suit the needs of individual patients. To achieve this goal, methods, cells, and materials are important components of the 3D bioprinting approach to precisely map tissue structures and manufacture new artificial organs in layers [16,17] both for in vitro and in vivo applications. 

### 2.1. Methods

Compared to traditional manufacturing technologies, 3D bioprinting enables the deposition and precise patterning of living cells and biological materials through a layer-by-layer fabrication approach [18]. Three-dimensional bioprinting techniques can be classified into distinct process categories [19]: material extrusion (mechanical/pneumatic), material jetting (inkjet [20], microvalve [21], laser-assisted [22] and acoustic [23]), and vat-photopolymerization (stereolithography (SLA) [24], digital light processing (DLP) [25], and two-photon polymerization (2PP) [26]). Each of these methods has specific strengths, weaknesses, and limitations (Table 1) [27], and the choice of a suitable bioprinting technique should depend on the intended application.

The bioprinting process consists of three distinct phases. To successfully manufacture precise anatomical shapes and exactly mimic the patient’s disease and/or defect requirements, it is necessary to start following several steps: (1)Pre-processing—the generation of digital models. Non-invasive imaging techniques, such as computed tomography (CT), magnetic resonance imaging (MRI), ultrasound (US), and optical coherence tomography (OCT), can be used to capture specific scanning data. Furthermore, 3D modeling software can help reconstruct 3D information from digital images. After an initial conversion to a standard tessellation language (STL) file to represent objects in the 3D cartesian coordinate system [28], the STL file is further processed to generate a .gcode file, the code necessary to interpret the digital model by the bioprinter [29,30].(2)Processing—the choice and preparation of the bioink, in addition to the bioprinting process itself. The selection of suitable bioink characteristics is made considering the functionality of the tissue of interest, its physical and chemical properties, in addition to the value of the source and the desired ability of the cells to include in the target environment [31,32].(3)Post-processing—stabilization and post-production maturation of the printed model [33]. This stage includes all steps after completion of the 3D bioprinting and before in vitro experimentation or implantation of the construct [34]. Indeed, also with the purpose of implanting the printed construct in vivo as organ replacement, post-processing may require a period of in vitro culture using different environmental conditions and signals, including stimulation strategies (mechanical, electrical, or electromechanical) to obtain mature and functional biocompatible substitutes [35].

### 2.2. Cells

Most human tissues contain cellular components, so cell integration and encapsulation are essential for the production of functional tissue constructs [17]. In fact, the chosen cell line determines the design and functionality of the final tissue construct [33,48]. One of the most widely used cell types for the generation of 3D tissue models is stem cells, as they can differentiate into a specialized cell type of interest while also continuously dividing and renewing [49,50]. Stem cells can be isolated from different sources, including embryonic and induced pluripotent stem cells or adult stem cells [51,52,53]. Mesenchymal stromal cells isolated from bone marrow, cord blood, or adipose tissue are often employed for the production of 3D tissue models, as they are able to easily differentiate in all mesoderm lineages, such as bone, cartilage, and fat [54,55]. However, different specialized progenitor cells have also been used so far for 3D bioprinting of specific tissues such as dermal fibroblasts for skin [56,57], or human chondrocytes for cartilage [58,59]. 

Clinically, for the generation of 3D implantable constructs, autologous cell sources are preferred because cells are derived from the same individual in which they will be used, reducing the risk of host rejection and graft versus host disease. However, allogeneic cells can be easily adopted for in vitro studies, allowing the production of different and heterogeneous constructs to study general cell behaviors or the effect of drugs and treatments. 

### 2.3. Materials 

3D printing materials are chosen according to the target application [60]. To exploit the potential of custom 3D bioprinting, hydrogels of different nature during the pre-gelation phase (i.e., before polymerization) can be used as bioinks [61]. Hydrogels are defined as 3D networks that comprise crosslinked hydrophilic polymer chains [62] and can be produced from a wide range of sources [63,64] to meet the specific requirements of each desired application [65]. In fact, each tissue of the human body has its own unique physical and mechanical properties [17], which have an impact on cellular functionality and therefore also on the choice of biomaterial. Furthermore, hydrogels have the ability to hold living cells, modify chemical structures, adjust biodegradation properties, and guarantee adequate resolution during printing [66].

Hydrogels are classified according to their source material and therefore can be grouped as natural and synthetic hydrogels [67]. The formers are primarily composed of natural materials such as agarose, alginate, chitosan, collagen, gelatin, fibrin/fibrinogen, hyaluronic acid (HA), and silk. On the contrary, synthetic hydrogels consist of synthetic materials such as polyurethane (PU), polyethylene glycol (PEG), polylactic acid (PLA), and polyvinyl alcohol (PVA). Natural materials have several advantages over synthetics [68,69], mainly related to the biomimicking of the composition and structure of human organs, self-assembling ability, biocompatibility, and biodegradation properties [66].

Among all the different types of natural bioink, those produced from the decellularized extracellular matrix (dECM) of human or animal organs have the potential to support specific cell types and trigger the innate regenerative process by providing a microenvironment closer to the native one [70]. The extracellular matrix (ECM) is a complex network of macromolecular substances produced and secreted by cells in tissues. ECM forms the skeleton of tissues and organs and directly influences cell behavior through specific receptors on the cell surface [70,71]. The bonds between cells and ECM allow cells to sense their surroundings and actively modulate their behavior [72]. The dECM of a tissue or organ is obtained using various physical and chemical methods, including detergents, freeze–thaw cycles, or enzyme agents [73,74]. These processes aim to remove all the original cellular components of the tissue while maintaining the structure and composition of the natural ECM. For these reasons, bioinks made of dECM can be considered suitable materials to generate 3D printed products, mimicking the complex structures and properties of each tissue, while retaining a specific functional composition [75,76,77,78,79,80].

## 3. Advantages of Naturally Derived Bioinks: Bioactivity and Biocompatibility

The main merit of hydrogels is the property of retaining large amounts of water [77]. The swollen state of the hydrogels is obtained by achieving an osmotic equilibrium given by the entry of water or aqueous biological fluids and by the cohesive forces exerted by the polymers that compose the biomaterial [81]. In addition, naturally derived hydrogels possess high biocompatibility with human tissues. The definition of biocompatibility is based on the ability of the biomaterial to support cellular activity, consent the transfer of physical and molecular information, and, finally, not injure tissues with toxic degradation products [82]. The biocompatibility of a bioink is intended as the ability to host living cells, allowing gas, nutrients and material exchange with the environment and possibly supporting cell proliferation, maturation, and surrounding remodeling to better suit cell needs. Naturally derived hydrogels are spontaneously endowed with biocompatibility and offer the possibility of selectively guiding cells toward physiological behaviors. Moreover, they have physiological rates of biodegradability that match the aptitude of cellular components to replace the material [83], generating nontoxic degradation products that are quickly cleaned or recycled by the tissues [84]. 

Natural hydrogels are composed of molecules that cells recognize as ‘self’ and physiological, and for which they possess natural receptors that not only allow the engraftment into the scaffold, but also the trigger of vital cellular signaling and molecular mechanisms. Among hydrogels of natural origin, those obtained directly from biological tissues, such as collagen, HA, fibrin, or ECM-derived hydrogels, offer great bioactive characteristics because they are normally present in the cellular environment. 

Collagen is the most prevalent protein in mammalian tissues, and collagen-based hydrogels are frequently used for biomedical applications [85,86,87,88]; they were used as both 3D scaffolds for in vitro oncological studies [89] and as cell or drug carriers in in vivo applications [90]. Koch et al. have printed a construct with the use of a laser-assisted bioprinter, to generate a bi-layered construct capable of replicating human dermis and epidermis [91]. Furthermore, Shi et al. [92] have printed six-layer cellular structures using an extrusion-based bioprinter. Unlike the work of Koch et al., Shi and colleagues used a mixture of methacrylated gelatin (GelMA) and collagen as ink material. In fact, collagen hydrogels are not often used as bioinks because of collagen mechanical instability and a slow gelation rate at physiological temperatures. These characteristics limit the possibility of the printed structure to maintain its shape and geometry [93].

HA is naturally present in the ECM of mammals and, as a tissue implant, it can be left in the body where it can dissolve or be absorbed. It also has the ability to maintain a hydrated environment, being an ideal material to promote wound healing and regenerate injured tissues [94]. Similarly to collagen, HA lacks mechanical integrity to function as an independent bioink and is frequently combined with other components [95]. For this reason, Zhang et al. [96] used the 3D extrusion bioprinting technique to generate a gelatin–fibrin–HA hydrogel layer to assess the formation of vascular networks and the vascular lumen.

Fibrin is another naturally occurring protein network that forms a temporary structure during physiological wound healing, and it is widely implemented in tissue engineering and cell culture applications because it can be polymerized into hydrogels. Among the different 3D bioprinting technologies, the most suitable for fibrin-based bioinks are jetting and extrusion-based techniques [97]. Although the mechanics of fibrin has been studied at various hierarchical scales, a deep understanding of this material remains incomplete for the correlation among fibrin fiber orientation, network structure, and mechanical response [98].

Decellularized ECM hydrogels retain the native structure and composition of ECM, and for this reason they have the ability to induce tissue-specific characteristics by choosing the preferred tissue source [99,100]. For example, ECM-derived hydrogels were frequently used as a supporting microenvironment for organoid culture [101]; Giobbe and coworkers [102] demonstrated the superiority of small intestine dECM-derived hydrogel in conditioning endoderm-derived organoid proliferation and maturation compared to the standard use of Matrigel^®^. In particular, dECM is one of the few natural biomaterials that is commonly used as a bioink on its own (Figure 1) and has been used to produce a variety of tissues, including the heart [103], skin [104], liver [105], intestine [106], bone [107], and skeletal muscle [108]. For a detailed overview of 3D bioprinting with dECM bioinks, see the review article by Chae et al. [109] in which the authors summarized various dECM-based bioink formulations and their tissue engineering applications. 

The formation of ECM-derived hydrogels is generally based on three necessary steps: (i) tissue decellularization [110], (ii) solubilization, digestion, and dissolution in acidic solution [111], and (iii) temperature and pH-controlled neutralization [35]. The last step is required to trigger spontaneous reformation of intramolecular bonds into a homogeneous gel. 

Using hydrogel bioinks derived from dECM for 3D printing allows the exploitation of the following advantages: Ability to maintain the same biological activity of the natural matrix [112]. dECM hydrogels retain numerous structural and soluble components found in native tissue, such as cell adhesion proteins, growth factors, and glycosaminoglycans. The presence of bioactive factors, such as cytokines, chemokines and growth factors, can enhance cell viability and proliferation. Indeed, it was shown that after addition of bioactive factors into bioinks, cell proliferation and ECM protein production increased compared to hydrogels without bioactive factors [113]. The characteristic of including a variety of structural proteins together with soluble factors and cytokines makes these types of hydrogels much more complete than other bioinks of natural origin. In addition, they support a constructive, site-appropriate remodeling response when implanted in a wide variety of anatomic sites [114,115,116].No immunogenic cell material due to decellularization. This prevents infection transmission and avoids an immune reaction, allowing the use of allogeneic or xenogeneic dECM [117].Injectability. The dECM pre-gel fluid can be extruded or injected directly into targeted areas or tissues using minimally invasive techniques [118,119] and can be induced to polymerize at physiological temperatures to form a hydrogel that perfectly fits the targeted organ, stimulating regeneration and ultimately serving as carrier of factors or molecules [120].

## 4. Challenges of Naturally Derived Bioinks: Mechanical Properties

To take advantage of all the potential offered by 3D bioprinting, it is essential that the appropriate materials preserve both bioactivity, to meet needs of cells, and biomechanical properties, to technically consent to the printing process [121,122] (Figure 2). 

A bioink must exhibit printability and shape fidelity features, which depend on specific mechanical properties [123]. The printability is determined by different parameters, such as the surface tension of the bioink during the printing process and the ability to crosslink on its own. Furthermore, the reliability of printing strictly depends on the viscosity of the bioink [124]; the fidelity of the printing generally increases with increasing viscosity. However, high viscosity implies an increase in shear stress and pressure required to properly extrude the material, a process that can be harmful to loaded cells [42,125]. Tirella et al. [126] studied the effect of bioink viscosity on cell viability and Kong et al. [127] obtained enhanced cell viability by encapsulating cells when low viscosity pre-gelled solutions were considered. Moreover, Ouyang and colleagues [128] studied the relationship between bioink rheological properties and embryonic stem cell viability during printing. They confirmed that a higher viscosity results in a lower cell viability and cell death or damage is due to induced shear stress during the extrusion process. In addition to this, deformation and collapse of printed constructs may easily occur when printing low-viscosity materials, while the extrusion bioprinter may jam when high-viscosity hydrogels are used [66,126]. Therefore, it is necessary to find a viscosity value that guarantees both printability and cell viability.

The shape stability of the printed construct also closely depends on the yield stress, a critical shear stress value below which a material acts as a solid and above which it flows like a liquid [129]. This is an important feature of bioinks, which will flow as liquid to be printed in a controlled manner. However, as for viscosity, higher yield stress requires increased extrusion pressures, which can negatively impact cell viability. 

Bioink mechanical requirements also include structural integrity, practicability, and resolution. Bioinks should provide enough strength and structural support until printed cells produce their own ECM components in the 3D architecture [130]. The mechanical support depends on the type and concentration of the polymer components. Although a low concentration is probably more supportive of cell viability, it would result in poor mechanical strength and can induce collapse of printed constructs or, in the worst case, could not be printable [131]. At the same time, an excessive polymer concentration could impede cell activities, so cells remain passive in the material [93]. This phenomenon is due to the role played by the stiffness of the printed substrate and the effect exerted on cell behavior [132]. In fact, tissue maturation is found to be highly stiffness dependent. Stiffness-tunable hydrogels can be obtained by changing the degree of gels [133,134,135], stimulating external conditions [136,137], changing the molecular weight of materials [138], modifying the proportions of components [139,140] and adding nanomaterials [141,142,143]. 

Stiffness and mechanical properties are also governed by the crosslinking process [144]. Crosslinking is an important aspect in preserving the shape of bioprinted constructs, thus minimizing structural collapse [42]. Collagen-based, dECM, and conventional natural hydrogels are randomly crosslinked, and single-network hydrogels are formed with no internal mechanism for mechanical energy dissipation. Although hydrogels used in tissue engineering are suitable for cell growth, they lack rapid solidification during the printing process, restricting the diversity of inks and further inhibiting their possible wide range of applications [145,146,147,148]. Adapting hydrogels to 3D bioprinting has proven to be a challenge. In fact, prior to crosslinking, pre-gels are typically liquid polymer solutions, which are hardly printable formulations that do not support the deposition of subsequent layers. This characteristic is due to a very slow crosslinking that maintains the hydrogel in a weak condition for a long time window. To overcome this challenge and support the bioink during printing, different solutions have emerged [149]. With the help of assistive materials, soft hydrogels may be printed into complex shapes with high fidelity. Assistive materials are used to provide only temporary help to the bioink. In fact, once the goal is reached, they are removed from the extruded structure. Assistive materials can have a dual function: they can serve as a supporting bath, into which the bioink is printed, or can be printed themselves (sacrificial inks). 

In the extrusion-based 3D printing strategy, pre-gel structures are printed within a fluid bath [145]. The solidification step can be performed during [150] or after printing [151,152]. The fluid bath can hold the extruded pre-gel structures in a liquid state for long periods of time; therefore, it is not necessary that bioinks possess rapid solidification properties. Shiwarski et al. [153] summarized the current achievements of the emerging 3D bioprinting method called freeform reversible embedding of suspended hydrogels (FRESH) 3D printing. Using this technology, the bioink can be extruded within a thermo-reversible support bath composed of a gelatin microparticle slurry that provides support during printing and is then melted at 37 °C [154,155,156]. Although this solution supports 3D soft hydrogel bioprinting, it is difficult to match the kinetics of gelatin dissolution and hydrogel crosslinking [150]. In addition, using a FRESH bath does not modify the stiffness of the printed constructs. 

Sacrificial inks in the 3D extrusion method have primarily been printed separately from the bioink to leave void spaces once removed. These materials are commonly used to generate internal hollows within a printed structure to mimic, for example, vasculature-like networks [149,157]. One of the most common sacrificial inks is Pluronic^®^ F-127, a triblock copolymer composed of polypropylene glycol (PPG) and PEG. It has been frequently used and printed due to its biocompatibility and the desirable conversion from gel to fluid with a temperature is reduced to 4 °C [66,158]. Recently, sacrificial inks have also been used as thickener components within soft hydrogels. In this case, the temporary addition of the sacrificial ink modifies the overall bioink characteristics, increasing the mechanical properties. In addition to many advantages, there are still some key challenges in the development and use of sacrificial inks, especially the risk of destruction of the printed structure during elimination of the supporting material [149]. 

Among hydrogels of natural origin, several have mechanical properties suitable for printability and shape fidelity, due to the possibility of modulating their composition to obtain the necessary viscosity. Collagen-based materials provide mechanical strength and allow structural organization of cell and tissue compartments [159]. However, collagen gelation is typically achieved using thermally driven self-assembly, which is difficult to control [154]. So far, dECM hydrogels have been used to print different types of tissues, but the resulting stiffness of the constructs is generally different from the physiological stiffness of the native tissue [160], therefore conditioning the behavior of the included cells, at least initially. Moreover, a lower stiffness could lead to failure of hydrogel implants at load bearing sites [161]. To ensure structural stability of naturally derived hydrogels, several strategies have been developed: incorporation of crosslinkers or additives, employing the addition of chemical modifications, and depositing fibers in set geometries [162,163,164]. 

To exploit all the advantages using natural hydrogels and best mimicking native tissues, the ultimate goal is to obtain 3D printed constructs with physiological stiffness and mechanical properties without compromising cell viability and maturation of the printed constructs.

## 5. Bioink Reinforcement and Crosslinking

As previously reported, hydrogel networks are conventionally strengthened by increasing their polymer content and crosslink density or by adding modifiers to the polymer solution [165,166]. However, increased polymer content, dense crosslinks or altering agents can interfere with cell viability by reducing the permeability and porosity of the material [146]. To meet different needs and properties of biomaterials, including natural hydrogels, diverse bioink reinforcements were developed (Figure 3). Specifically, several trigger conditions have been used to prepare advanced hydrogels for adaptation to 3D printing [84,167]. Crosslinking processes are broadly classified into physical and chemical methods, based on the mechanism of action [168], and result in a crosslinked polymer network reversibly or irreversibly, respectively [64,169].

In physically triggered gels, crosslinking occurs via secondary forces such as ionic/electrostatic interactions, hydrophobic/hydrophilic interactions, polymerized entanglements, hydrogen bonds, crystallization/stereocomplex formation, metal coordination and π–π stacking or van der Walls forces [167,170,171,172]. In chemically formed structures, instead, covalent or coordinate bonds between polymer chains produce a stable hydrogel network, using molecules or ionic crosslinking agents [170,173]. 

### 5.1. Physical Crosslinking

Physically crosslinked hydrogels are formed as a result of the physical crosslinking interactions. The prominent advantage of a physical crosslinking is biomedical safety owing to the absence of chemical agents [171], consequently avoiding potential cytotoxicity from unreacted chemical crosslinkers [167,173]. This gelation process is generally reversible, and, more importantly, this class of hydrogels is stimuli-responsive with self-healing and injectable properties at room temperature [167,174]. Therefore, they can be designed as bioactive hydrogels for drug delivery and encapsulation of living cells [175].

#### 5.1.1. Temperature-Triggered Hydrogels 

Thermal crosslinking through heating or cooling of the natural polymer solution is one of the simplest curing methods, and it can be applied to polymers that can sustain heating or cooling during the 3D bioprinting process [176], such as gelatin, elastin, agarose, and collagen [177]. Thermal condensation is the result of self-assembly and aggregation between polymer chains in an aqueous solution, promoting the transition from a dispersed micelle state to a dense 3D network structure. Phase separation occurs when the polymer solution is above or below a specific temperature, called the critical dissolution temperature (CST) [178]. Transition temperatures are defined as the upper CST (UCST) and lower CST (LCST). Above UCST, the material will dissolve and the phase transition of the thermally responsive polymer will occur in the environment below this temperature [179]. On the contrary, a thermosensitive hydrogel can be dissolved at a low temperature. In this situation, the phase transition occurs in an environment above LCST. When the solution is heated above the LCST, the molecules precipitate from the solution and undergo a sol–gel phase transition [174,180]. In all of these cases, the variation in temperature may cause a change in intermolecular forces between the hydrogels (swelling and deswelling) [181]. Generally, the gelation time in thermal crosslinking is longer than that of other curing methods. Furthermore, in the temperature-triggered hydrogels, the degree of crosslinking cannot be precisely controlled. Temperature-sensitive hydrogels have been widely studied as controlled drug delivery systems, where loaded agents can be precisely released at the desired temperature [181].

#### 5.1.2. pH-Sensitive Hydrogels 

The pH-sensitive hydrogels change their volumes in response to a change in the pH of their environment. More specifically, pH-sensitive hydrogels consist of a polymer containing weak acidic or basic groups that become more ionized in a higher or lower pH environment, respectively [182]. Ionization in the form of protonation or deprotonation alters the electrostatic force between polymer chains, which causes volume changes in hydrogels. Cationic hydrogels swell at low pH (acidic condition), while anionic hydrogels swell at higher pH (basic condition) [181,183,184]. For both types of hydrogels, an ion concentration gradient between inside and outside of the gel is produced accordingly when the environmental pH is changed [185]. This gradient causes the penetration of mobile ions across the hydrogel, which induces the modification of the osmotic pressure on the surface of the hydrogel, therefore resulting in a volume change. An osmotic driving force, opposite to the crosslinks, allows additional free water to enter the hydrogel and reach swelling equilibrium through the elastic restoring force [181,186]. Chitosan is an example of an existing natural cationic polymer frequently used for fabricating pH-sensitive hydrogels, because of its protonatable amine groups. 

#### 5.1.3. Ion-Responsive Hydrogels

Another physical approach used for hydrogel crosslinking is based on the application of an ionic mechanism [187]. It usually involves two molecules of opposite electric charges to induce gelation. In this rapid and extensively applied crosslinking technique, hydrogels can be formed under mild conditions at room temperature and physiological pH [176]. For example, alginate, a naturally derived polysaccharide with residues of mannuronic and glucuronic acid, can form a 3D gel structure by exploiting the ionic interaction mechanism. In fact, divalent cations, such as calcium (Ca^2+^), barium (Ba^2+^), and magnesium (Mg^2+^) [167], can only bind to guluronate blocks with a high degree of coordination. Subsequently, adjacent polymer chains can form junctions between guluronate blocks, resulting in a three-dimensional structure [167,188]. 

#### 5.1.4. Light-Responsive Hydrogels

Another approach to trigger polymerization is via illumination with a specific wavelength. This light-driven method offers several advantages: rapid formation of hydrogel networks at room or physiological temperature, tunable mechanical properties, the potential to use natural sunlight [189], and an accurate selection of the crosslinking site. In fact, in light-activated crosslinking, photoinitiated polymerization takes place under light exposure and only irradiated areas are involved in the crosslinking process [167,190]. Photosensitive hydrogels can change their volume under short exposure to visible or ultraviolet (UV) light in the presence of light-sensitive compounds, called photoinitiators [191]. Photoinitiators allow for the formation of covalent bonds, participating in the generation of a chemical crosslinking. Always using light as a curing agent, hydrogels can also be crosslinked through physical interactions under three different approaches [181]:Photosensitive hydrogels can absorb and emit light as energy. Light can be converted into heat through photosensitive moieties to trigger the polymer phase transition temperature and the consequent polymerization. This approach occurs in a similar way to temperature-sensitive hydrogels [192].Photosensitive molecules can be ionized through light irradiation to produce ion-sensitive hydrogels or crosslinking induced by variation in ionic concentration.Chromophoric groups can be incorporated into the hydrogel matrix to alter physical properties (geometry, dipole moments) under light irradiation. This method can facilitate the formation of hydrogels after in vivo injection, which is attractive for drug delivery and tissue engineering [193].

### 5.2. Chemical Crosslinking

Different to the physical approach, chemical reticulation always requires precise and controlled process conditions, such as to allow the development of more accurate and hierarchically complex microenvironments [174]. In fact, the most stable and tunable hydrogels can be obtained through chemical crosslinking. Until now, different chemical crosslinking mechanisms have been reported to form covalent bonds among modified polymer chains in hydrogel systems and they involve small crosslinking molecules, photo- and enzymatic-induced curing [194]. 

#### 5.2.1. Small Molecule Crosslinking Agents

Incorporation of specific small crosslinking agents, including glutaraldehyde (GA), dopamine, carbodiimide, citric acid, and tannic acid is traditionally considered an effective way to simultaneously tailor the mechanical properties and functionality of hydrogels [194,195]. 

Among them, GA has been extensively used as a chemical crosslinker to polymerize various types of hydrogels, including natural ones. Its main characteristic is to significantly improve the mechanical properties and durability of the hydrogel [196]. GA reacts with the amine or hydroxyl functional groups of proteins and polymers through a Schiff base reaction and connects the biopolymeric chains via intramolecular or intermolecular bonds. Therefore, all free amine groups that are present in the protein structure react with GA, forming a strong crosslinked network [195]. For many years, GA has been used as the gold standard curing method [197], but its application is to the date restricted due to toxic side effect on cells and tissues. In fact, the functional aldehyde groups of GA cause severe inflammation and the application of GA in commercial products was limited [198]. As an alternative, dopamine, caffeic acid, tanning acid, and carbodiimide agents have attracted much more interest and are frequently introduced into polymer networks to improve the performance of hydrogels [194].

#### 5.2.2. Free Radical Polymerization Crosslinking

Free radical polymerization can convert linear polymers into 3D polymer networks. It commonly uses free radicals generated by initiators to induce the formation of new free radicals on linear polymers under specific conditions of temperature, pH, or radiation, and induces the polymerization process through the coupling of new free radicals [199]. Photopolymerization has frequently been used as the main curing method. Photoinitiator molecules, or UV or visible light can be used to trigger polymerization of materials containing unsaturated bonds to form hydrogels [200]. This chemical approach provides some advantages, such as mild reaction conditions, high structural ability, and tunable mechanical properties [201]. Moreover, this method allows remote manipulation without introducing additional crosslinkers and therefore prevents by-product generation [181,202]. Furthermore, the crosslinking density and physicochemical properties of photocrosslinkable hydrogels can be precisely controlled by adjusting the intensity of light and the exposure time to promote cell proliferation and differentiation [203,204,205]. Among the 3D bioprinting methods of photocrosslinkable constructs, that of free radical polymerization of methacrylate-based monomers is the most frequently used. The chain growth polymerization is initiated via photoirradiation, which produce free radicals by dissociating photoinitiators, subsequently added to the bioink. Then, the radicals produced can react with the functional groups of the polymers and bind them together to form 3D network structures [176]. 

To enhance the structural integrity and stability of natural bioinks, the photocrosslinkable process using the methacrylate reaction has been recently proposed in various hydrogels, such as GelMA, methacrylated HA, and methacrylated collagen [206,207,208]. Moreover, Kim et al. [209] used dECM methacrylate (dECM-MA) derived from porcine skeletal muscles as a bioink to produce muscle-like 3D tissue. The methacrylate was combined with fibrillated PVA to fabricate a uniaxially orientated dECM-MA-patterned structure.

#### 5.2.3. Enzymatically Crosslinked Hydrogels

Enzymatic crosslinking is an attractive method, as it offers the possibility of kinetic manipulation of in situ gel formation by controlling enzyme concentration [210]. Enzymes can be employed as catalysts to promote the formation of covalent bonds between protein-based polymers. Catalyzed reactions occur at a neutral pH in an aqueous environment at moderate temperatures, as well as under normal physiological conditions in the human body [181]. The majority of enzymes involved in crosslinking are common enzymes that catalyze naturally occurring reactions [211,212,213]. So far, there are many types of enzymatically crosslinked methods for in situ hydrogel formation. For example, transglutaminase (TG) is a widely used enzyme catalyst that provides mild reaction conditions, fast gelation, and high cytocompatibility [214]. Moreover, double-network hydrogels can be generated by exploiting TG crosslinking in combination with other reactions. For example, Chen et al. produced a cytocompatible interpenetrating network hydrogel for cell culture and 3D bioprinting, using TG to reticulate gelatin in combination with alginate/Ca^2+^ [215,216]. 

### 5.3. Natural Crosslinkers

Although there is a wide variety of options for hydrogel crosslinking, not all of them are able to produce final products of suitable stiffness due to their difficult management and limited precision. For this reason, chemical crosslinking is the most used method for improving hydrogel stability. However, cytotoxicity associated with the chemical crosslinking is the major disadvantage of this method. To overcome these issues and obtain hydrogels that keep the cell environment as natural as possible, natural crosslinking agents have emerged. Natural crosslinkers not only improve the mechanical stability of hydrogels, but also guarantee their biocompatibility with biological systems.

#### 5.3.1. Genipin

Genipin (GP) is one of the most investigated natural crosslinkers because of its biocompatibility, biodegradability, and low cytotoxicity. It is a hydrolytic product extracted from the fruit of *Gardenia jasminoides* Ellis [217]. The fruit is an oriental folk medicine, used as an active ingredient in traditional Chinese medicine. GP reacts with materials containing primary amine groups, such as collagen, chitosan, gelatin, proteins, and dECM, to form covalently crosslinked networks. The crosslinking process occurs through a series of reactions that involve different sites on the GP molecule, which ends with a radical polymerization responsible for the blue pigment of the final product [218].

Many in vitro studies have revealed that the cytotoxicity of GP is significantly lower than that of GA, the molecule most commonly used molecule for chemical crosslinking [219,220]. Using the MTT assay with mouse fibroblasts, Sung et al. [219] demonstrated that GP is approximately 10,000 times less cytotoxic than GA and it can form stable crosslinked products with resistance to enzyme degradation in a manner similar to that of GA-fixed tissues. Furthermore, with a colony forming assay, it was suggested that cell proliferation after exposure to GP was approximately 5000 times greater than that observed after GA treatment [195,221]. 

Taking advantage of the abundant presence of collagen, Boso et al. [222] used GP to crosslink porcine diaphragm dECM hydrogels used as tissue patches for the treatment of diaphragmatic malformations. After crosslinking, they verified that hydrogels appeared to be unaffected by enzymatic degradation, suggesting potential resistance when used as in vivo tissue substitutes. Moreover, crosslinked hydrogels presented a densely packed inner architecture and collagen packaging, which made them suitable for subsequent mechanical stimulation.

#### 5.3.2. Proanthocyanidin

Proanthocyanidin (PA) compounds are naturally occurring plant metabolites widely available in fruits, vegetables, nuts, seeds, flowers, and barks [223]. PA belongs to the category known as condensed tannins, which consist of highly hydroxylated structures capable of forming insoluble complexes with carbohydrates and proteins [224]. 

PA was selected as a natural crosslinking agent to reticulate biopolymers in biological tissues. Han et al. [223] have investigated the cytotoxicity, crosslinking rate, and biocompatibility of PA as a collagen scaffold fixative. The results of these studies indicate that PA can efficiently crosslink collagen. Furthermore, PA is about 120 times less cytotoxic than GA and crosslinked matrices encourage cell ingrowth and proliferation [225]. Unlike fresh tissues, PA crosslinked structures showed stability comparable to that of GA-treated tissues after subcutaneous implantation in animal models. Therefore, PA crosslinked collagen matrices could be useful for designing tissue engineering scaffolds. In another study, Liu [226] selected PA as a crosslinking reagent to prepare a gelatin conduit for peripheral nerve regeneration. Reticulation of the gelatin conduit with PA improved resistance to enzymatic degradation and proved to be beneficial in enhancing cell adhesion, viability, and growth.

#### 5.3.3. Vitamin B2

Vitamin B2 (VB2, also called riboflavin) is a yellow edible water-soluble vitamin generated by plants and many microorganisms [227]. VB2 acts as a natural photosensitizing agent with complex photochemistry and is often used as a biocompatible photocuring agent to promote the formation of chemical crosslinks in 3D hydrogel networks. 

Riboflavin is widely used in ophthalmic applications to enhance stroma strength through UV irradiation in a completely non-toxic manner [228]. In fact, VB2 and UVA irradiation increases corneal rigidity as a result of covalent crosslinking of stromal collagens and core proteoglycan proteins [229]. Inspired by this VB2/UVA medical procedure, Jang et al. [230] incorporated riboflavin into heart dECM bioinks to improve extrusion during 3D bioprinting and achieve mechanical stiffness close to that of cardiac tissue.

## 6. 3D Printed Tissue Cultures and Future Perspectives

Although 3D printing of complex organs has not yet been possible, a first clinically relevant step through the use of biological ink for the production of implantable constructs has already been taken. By utilizing CT imaging, image segmentation, dECM-derived hydrogel, and the FRESH printing process, Behre et al. [231] generated and implanted large scaffolds that precisely matched the geometry of recipient skeletal muscle defects, confirming the idea that structures obtained with bioprinted dECM can be successfully tailored to each individual patient’s needs. 

Regardless of the type of naturally derived inks and the mechanical modification that are implemented, currently the most investigated area is that of in vitro validation of these printed constructs, the analysis of cellular behaviors within the 3D environment, and the co-culture of different cell types to obtain, at least on the bench, a relatively complex tissue that demonstrates typical characteristics and functionality of the target organ [209,232]. This type of investigation is obviously preparative for future in vivo clinical applications, but it is also necessary to obtain reliable 3D models for drug screening. Through 3D bioprinting of decellularized porcine tongue, and head and neck squamous cell carcinoma, Kort-Mascort et al. [233] obtained the formation of tumor-like spheroids that display phenotypes previously reported in tumors of the oral cavity, and drug testing experiments demonstrated the reliability of using this platform for drug screening and personalized medicine.

## 7. Conclusions

Three-dimensional bioprinting is the most innovative tissue engineering approach to obtain both implantable constructs for organ replacement and in vitro tissue-like structures for disease modeling. Three-dimensional bioprinting performances are largely based on the bioink’s ability to produce stable high-resolution structures while maintaining cell viability during and after fabrication. Natural hydrogels, especially dECM-derived hydrogels, are suitable bioinks because they mirror the native environment in terms of structural and nonstructural protein composition. Many aspects concerning the mechanical properties of these biomaterials still need to be improved to enhance these products in the clinic, especially those related to reliable and nontoxic/safe crosslinking. In this sense, natural crosslinkers offer great advantages, because they can increase the stiffness and mechanical properties of printed bioinks without introducing into the natural microenvironment products or reagents that would have a negative impact on cell viability. To date, few preclinical works have used natural molecules for hydrogel crosslinking, and the range of available molecules is still limited. In the future, it will be necessary both to determine new agents of natural origin that are able to act as crosslinkers, and to optimize and standardize the use of this type of molecules to allow the safe manufacturing of 3D organs and tissues. These solutions may then be easily exploited to generate functional constructs that can be translated into the long-awaited clinical practice.

## Figures and Tables

**Figure 1 biomedicines-11-01742-f001:**
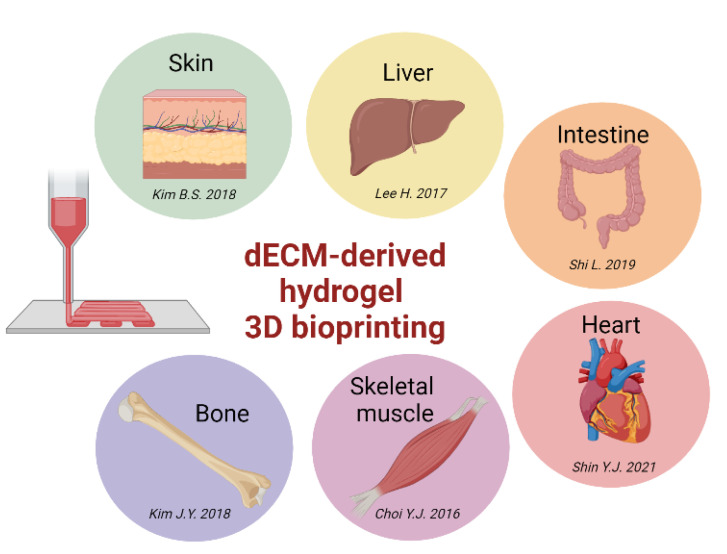
Graphical representation of different tissues produced by 3D printing of dECM-derived hydrogels. Image created with BioRender© [103,104,105,106,107,108].

**Figure 2 biomedicines-11-01742-f002:**
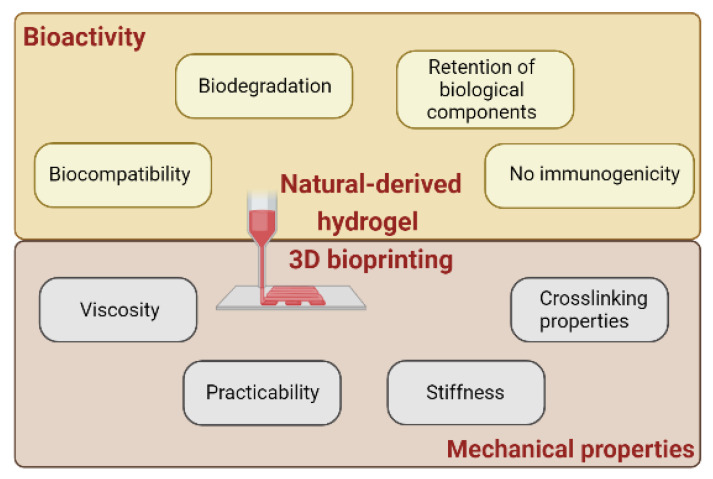
Graphical representation of the biological and mechanical properties that naturally derived hydrogels must exhibit to be efficiently used as bioink for 3D printing purposes. Image created with BioRender©.

**Figure 3 biomedicines-11-01742-f003:**
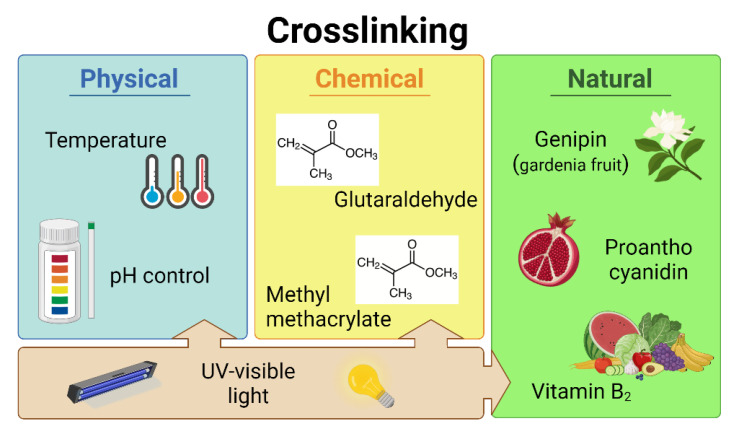
Schematic of the main curing methods commonly used in hydrogel crosslinking grouped as physical, chemical, or natural agents. Image created with BioRender©.

**Table 1 biomedicines-11-01742-t001:** List of strengths and limitations of the three main printing technologies for biomedical applications.

	PROs	CONs	References
Extrusion-based	Good quality of vertical structure; chemical, photocrosslinking; shear thinning and temperature gelation method; microscale resolution; high cell density; piston-, pneumatic-, or screw-driven.	Slow print speed; poor cell viability (40–80%) due to shear damage; low resolution.	[27,28,36,37,38]
Jetting-based	Low cost; high resolution; fast printing speed; chemical and photocrosslinking gelation method; thermal-, electrostatic-, laser-pulse or piezoelectric-driven.	Narrow ranges of printable biomaterial viscosities; high probability of cell damage, and cell lysis; non-uniform droplet size; nozzle clogging risk.	[9,27,39,40,41,42,43,44,45]
Vat photo polymerization	High resolution and fabrication accuracy, high production speed, dimensional stability, fast processing.	Limited choice of biocompatible materials, high cost, time- and energy-intensive.	[46,47]

## Data Availability

Not applicable.

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
