# Peer review of "Three-Dimensional Bioprinting of Naturally Derived Hydrogels for the Production of Biomimetic Living Tissues: Benefits and Challenges"

_biomedicines, 2023, doi:10.3390/biomedicines11061742_

Round 1
Reviewer 1 Report
1. In the Introduction, what are the features and innovations of the current article compared to published articles in similar fields?
2. Background descriptions for hydrogel crosslinking and preparation can be strengthened by citing 10.1016/j.jclepro.2021.128221; 10.1016/j.reactfunctpolym.2020.104501 and what are the advantages of the current work compared to published articles?
3. At the end of the article, the author should show a more critical and own viewpoint: current weaknesses and future prospects. This way, the reader can get more points of view in a simple and clear way.
4. The Conclusion section should be strengthened: the important results and main conclusions drawn in this paper should be highlighted and presented in more precise language.
5. There are some formatting errors in the article. For example, spelling of references must be checked to meet the journal style (such as Reference 28). Please check carefully and use abbreviation properly.
No
Author Response
Comments:
- In the Introduction, what are the features and innovations of the current article compared to published articles in similar fields?
We thank the Reviewer for their suggestion. We modified final sentences of the introduction as follows (lines 71-77):
“In this review, we focus on benefits and challenges of using natural-derived hydrogels as bioinks for 3D printing approaches and tissue-like construct production, specifically highlighting their biological and mechanical characteristics. Given the wide range of available biological materials and the different properties of each individual biomaterial, we summarize the most popular physical, chemical and natural crosslinking options to modify and tune the properties of natural-derived hydrogels to better suit 3D bioprinting process.”
- Background descriptions for hydrogel crosslinking and preparation can be strengthened by citing 10.1016/j.jclepro.2021.128221; 10.1016/j.reactfunctpolym.2020.104501 and what are the advantages of the current work compared to published articles?
We are sorry that we cannot follow the Reviewer's suggestion, but we do not find a direct relationship between the indicated publication (Recent advances in polysaccharide-based adsorbents for wastewater treatment. Journal of Cleaner Production, DOI:10.1016/j.jclepro.2021.128221) and the topic under discussion.
The second suggested article (Recent advances in natural polymer-based drug delivery systems. Reactive and Functional Polymers, DOI: 10.1016/j.reactfunctpolym.2020.104501) was added as reference n. 88.
- At the end of the article, the author should show a more critical and own viewpoint: current weaknesses and future prospects. This way, the reader can get more points of view in a simple and clear way.
- The Conclusion section should be strengthened: the important results and main conclusions drawn in this paper should be highlighted and presented in more precise language.
Following the suggestions 3 and 4, we added a new section highlighting future perspectives (6. 3D printed tissue cultures and future perspectives, lines 602-621), and some limitation in the conclusion section (lines 634-640):
“To date, few preclinical works have used natural molecules for hydrogel crosslinking, and the range of available molecules is still limited. In the future, it will be necessary both to determine new agents of natural origin that are able to act as crosslinkers, and to optimize and standardize the use of this type of molecules to allow a safe manufacturing of 3D organs and tissues. These solutions may then be easily exploited to generate functional constructs that can be translated into the long-awaited clinical practice.”
- There are some formatting errors in the article. For example, spelling of references must be checked to meet the journal style (such as Reference 28). Please check carefully and use abbreviation properly.
Thank you for the advice, we checked carefully and edited the style properly.
Reviewer 2 Report
Comments:
1. It is recommended that the authors include some relevant references for the following sentence (Line 36 – 40) that highlight the benefits of 3D bioprinting of organs/tissues – (i) defined shape, size and geometry, (ii) distribution and positioning of biomaterials, cells etc, (iii) cell-cell communication. The authors can refer to some of the following papers below:
a. "Resolution and shape in bioprinting: Strategizing towards complex tissue and organ printing." Applied Physics Reviews 6, no. 1 (2019): 011307.
b. "3D bioprinting of tissues and organs." Nature biotechnology 32, no. 8 (2014): 773-785.
2. The standard ASTM classification of 3D bioprinting techniques should be used in Section 2.1 – extrusion-based, jetting-based and vat photopolymerization-based bioprinting with relevant references. Please modify Table 1 accordingly.
a. Extrusion-based
i. "Current advances and future perspectives in extrusion-based bioprinting." Biomaterials 76 (2016): 321-343.
b. Jetting-based
i. "Improving printability of hydrogel-based bio-inks for thermal inkjet bioprinting applications via saponification and heat treatment processes." Journal of Materials Chemistry B 10, no. 31 (2022): 5989-6000.
c. Vat photopolymerization
i. "Vat polymerization-based bioprinting—Process, materials, applications and regulatory challenges." Biofabrication 12, no. 2 (2020): 022001.
3. As the focus of this paper is on 3D bioprinting of natural derived hydrogels, the authors should perform an analysis of publication trends for the different types of natural hydrogels used in bioprinting (collagen, gelatin, alginate, dECM etc) and discuss the top 3/top 5 commonly used hydrogels in individual sections instead of discussing the different natural bioinks together (which can be confusing to readers as they might not be able to differentiate what are the crosslinking methods/crosslinkers for each type of bioinks). The authors can consider reorganizing the content as follows for more clarity:
a. Collagen/gelatin-based/alginate/dECM etc
i. Compatible with which printing techniques (extrusion, jetting, vat photopolymerization) at what concentration
ii. Printing strategy/crosslinking strategies
iii. Bioactivity/biocompatibility
iv. Mechanical properties
4. What is the outlook for 3D bioprinting of natural hydrogels for tissues/organs?
a. What are the recent works (within last 2 years) to achieve high cell viability in bioprinting?
b. How to accelerate the tissue maturation process?
c. The use of deep learning in bioprinting?
Author Response
Comments:
- It is recommended that the authors include some relevant references for the following sentence (Line 36 – 40) that highlight the benefits of 3D bioprinting of organs/tissues – (i) defined shape, size and geometry, (ii) distribution and positioning of biomaterials, cells etc, (iii) cell-cell communication. The authors can refer to some of the following papers below:
- "Resolution and shape in bioprinting: Strategizing towards complex tissue and organ printing." Applied Physics Reviews 6, no. 1 (2019): 011307.
- "3D bioprinting of tissues and organs." Nature biotechnology 32, no. 8 (2014): 773-785.
The Authors thank the Reviewer for their suggestion. We added the indicated references at lines 38 and 41, that are n.8 and n.9, respectively.
- The standard ASTM classification of 3D bioprinting techniques should be used in Section 2.1 – extrusion-based, jetting-based and vat photopolymerization-based bioprinting with relevant references. Please modify Table 1 accordingly.
- Extrusion-based
- "Current advances and future perspectives in extrusion-based bioprinting." Biomaterials 76 (2016): 321-343.
- Jetting-based
- "Improving printability of hydrogel-based bio-inks for thermal inkjet bioprinting applications via saponification and heat treatment processes." Journal of Materials Chemistry B 10, no. 31 (2022): 5989-6000.
- Vat photopolymerization
- "Vat polymerization-based bioprinting—Process, materials, applications and regulatory challenges." Biofabrication 12, no. 2 (2020): 022001.
- Extrusion-based
We modified Table 1 and edited the Section 2.1 according to the suggestion, adding a more precise definition of the three printing processes.
- As the focus of this paper is on 3D bioprinting of natural derived hydrogels, the authors should perform an analysis of publication trends for the different types of natural hydrogels used in bioprinting (collagen, gelatin, alginate, dECM etc) and discuss the top 3/top 5 commonly used hydrogels in individual sections instead of discussing the different natural bioinks together (which can be confusing to readers as they might not be able to differentiate what are the crosslinking methods/crosslinkers for each type of bioinks). The authors can consider reorganizing the content as follows for more clarity:
- Collagen/gelatin-based/alginate/dECM etc
- Compatible with which printing techniques (extrusion, jetting, vat photopolymerization) at what concentration
- Printing strategy/crosslinking strategies
- Bioactivity/biocompatibility
- Mechanical properties
- Collagen/gelatin-based/alginate/dECM etc
We appreciate the Reviewer's suggestion to list every single natural-derived hydrogel indicating characteristics and applications for each one. However, we believe that in literature there are already several review articles proposing this type of organization:
- Janitha M. Unagolla, Ambalangodage C. Jayasuriya, Hydrogel-based 3D bioprinting: A comprehensive review on cell-laden hydrogels, bioink formulations, and future perspectives, Applied Materials Today, Volume 18, 2020, 100479, ISSN 2352-9407, https://doi.org/10.1016/j.apmt.2019.100479.
- Sania Raees, Faheem Ullah, Fatima Javed, Hazizan Md. Akil, Muhammad Jadoon Khan, Muhammad Safdar, Israf Ud Din, Mshari A. Alotaibi, Abdulrahman I. Alharthi, M. Afroz Bakht, Akil Ahmad, Amal A. Nassar, Classification, processing, and applications of bioink and 3D bioprinting: A detailed review, International Journal of Biological Macromolecules, Volume 232, 2023, 123476, ISSN 0141-8130, https://doi.org/10.1016/j.ijbiomac.2023.123476.
- Xiang, Y., Miller, K., Guan, J. et al. 3D bioprinting of complex tissues in vitro: state-of-the-art and future perspectives. Arch Toxicol 96, 691–710 (2022). https://doi.org/10.1007/s00204-021-03212-y
- Benwood, C.; Chrenek, J.; Kirsch, R.L.; Masri, N.Z.; Richards, H.; Teetzen, K.; Willerth, S.M. Natural Biomaterials and Their Use as Bioinks for Printing Tissues. Bioengineering 2021, 8, 27. https://doi.org/10.3390/bioengineering8020027
and the addition of another one would not enrich the reader's point of view. Instead, we think it might be helpful to have a more general vision, in which do not look at the single material, but considering the characteristics that these materials must have to obtain reliable printed structures. In fact, we opted to divide the chapters on the basis of advantages and challenges that these natural derived hydrogels pose when they are chosen as bioinks.
However, following the advice of the Reviewer, we added more information for each mentioned biomaterial:
“Koch et al. have printed a construct with the use of a laser-assisted bioprinter, to generate a bi-layered construct capable of replicating human dermis and epidermis [91]. Furthermore, Shi et al. [92] have printed six-layer cellular structures using an extrusion-based bioprinter. Unlike the work of Koch et al., Shi and colleagues used a mixture of methacrylated gelatin (GelMA) and collagen as ink material. In fact, collagen hydrogels are not often used as bioinks because of collagen mechanical instability and a slow gelation rate at physiological temperature. These characteristics limit the possibility of the printed structure to maintain its shape and geometry [93].” – lines 200-207
“For this reason, Zhang et al. [96] used the 3D extrusion bioprinting technique to generate a gelatin-fibrin-HA hydrogel layer to assess the formation of vascular networks and the vascular lumen.” – lines 212-215
“For a detailed overview of 3D bioprinting with dECM bioinks, see the review article of Chae et al. [109] in which the Authors summarized various dECM-based bioink formulations and their tissue engineering applications.” – lines 233-235
- What is the outlook for 3D bioprinting of natural hydrogels for tissues/organs?
- What are the recent works (within last 2 years) to achieve high cell viability in bioprinting?
- How to accelerate the tissue maturation process?
- The use of deep learning in bioprinting?
With the attempt to meet the reviewer's suggestion, we added a chapter that aims to analyse the future perspectives of this type of technology applied both to 3D tissue culture and disease modelling, and to in vivo application (lines 602-620):
“6. 3D printed tissue cultures and future perspectives
Although 3D printing of complex organs has not yet been possible, a first clinically relevant step through the use of biological ink for the production of implantable constructs has already been taken. By utilizing CT imaging, image segmentation, dECM-derived hydrogel, and the FRESH printing process, Behre et al. [232] generated and implanted large scaffolds that precisely matched the geometry of recipient skeletal muscle defects, confirming the idea that structures obtained with bioprinted dECM can be successfully tailored to each individual patient's needs.
Regardless of the type of natural-derived inks and the mechanical modification that are implemented, currently the most investigated area is that of in vitro validation of these printed constructs, the analysis of cellular behaviors within the 3D environment, and the co-culture of different cell types to obtain, at least on the bench, a relatively complex tissue that demonstrates typical characteristics and functionality of the target organ [210,233]. This type of investigation is obviously preparative to future in vivo clinical applications, but it is also necessary to obtain reliable 3D models for drug screening. Through 3D bioprinting of decellularized porcine tong and head and neck squamous cell carcinoma, Kort-Mascort et al. [234] obtained the formation of tumor-like spheroids that display phonotypes previously reported in tumors of the oral cavity, and drug testing experiments demonstrated the reliability of using this platform for drug screening and personalized medicine.”
Reviewer 3 Report
This manuscript is a narrative review of methods devised to optimize bioinks derived from natural sources. Chemical modifications that seek to reinforce natural-derived bioinks are well described. In contrast, the introductory sections (1 and 2) need a thorough revision to bring the paper in the conceptual framework of the field. The major weak points of this text are the following:
1. The language of the manuscript needs refinements to be consistent with the evolving professional jargon of 3D bioprinting and biofabrication. For example, the definition of 3D bioprinting given on lines 33 and 58 deviates from the literature (see refs. [1,2] below). I would encourage the authors to read, the following papers by leading experts (I'm not a coauthor of these works):
[1] Guillemot F, Mironov V, Nakamura M. Bioprinting is coming of age: Report from the International Conference on Bioprinting and Biofabrication in Bordeaux (3B'09). Biofabrication. 2010;2(1):010201.
[2] Groll J, Boland T, Blunk T, Burdick JA, Cho D-W, Dalton PD, et al. Biofabrication: reappraising the definition of an evolving field. Biofabrication. 2016;8(1):013001.
[3] Groll J, Burdick JA, Cho DW, Derby B, Gelinsky M, Heilshorn SC, et al. A definition of bioinks and their distinction from biomaterial inks. Biofabrication. 2018;11(1):013001.
[4] Moroni L, Boland T, Burdick JA, De Maria C, Derby B, Forgacs G, et al. Biofabrication: A Guide to Technology and Terminology. Trends in Biotechnology. 2018;36(4):384-402.
2. The perspectives of 3D bioprinting, in my opinion, are overstated by the authors. In particular, the sentence "in a few years will allow the construction of patient-specific organs useful for transplantation purposes" is way more optimistic than the current literature:
[5] Sun W, Starly B, Daly AC, Burdick JA, Groll J, Skeldon G, et al. The bioprinting roadmap. Biofabrication. 2020;12(2):022002.
[6] Ng WL, Chua CK, Shen Y-F. Print Me An Organ! Why We Are Not There Yet. Progress in Polymer Science. 2019;97:101145.
3. Section 2, focusing on merely three bioprinting techniques, represents a 10-years-old view. Please revise it to provide a state-of-the-art picture of the field. Good references in this respect are [5] and [7]:
[7] Ji S, Guvendiren M. Complex 3D bioprinting methods. APL Bioengineering. 2021;5(1):011508.
4. In sections 3-5, I would cite and discuss the recent review by Prof. Cho's group:
[8] Chae S, Cho D-W. Three-dimensional bioprinting with decellularized extracellular matrix-based bioinks in translational regenerative medicine. MRS Bulletin. 2022.
5. To boost the attractivity of this review, please include at least one multi-panel figure in Section 3 to illustrate current applications of dECM-derived bioinks. As it stands, the paper is poorly illustrated. Approval to reproduce original figures from, e.g., refs. 86-93 can be obtained online via RightsLink (click on links such as "Request Permissions" or "Get Rights and Content" next to the abstract of the chosen paper).
Minor remarks and revisions, listed in the format <original text> => <proposed revision>
Line 35: most ideal => ideal
Line 36: Several are the possible benefits of this organ and tissue manufacturing: => Benefits of this organ and tissue manufacturing include:
Line 52: might result opposite => might differ
Line 136: What do the authors mean by "good resolution"? Please be more specific.
Line 151: secreted and synthesized => secreted
Line 162: Natural-derived bioink advantages: => Advantages of natural-derived bioinks:
Line 216: According to [3], the term "bioink" refers to a cell-laden bioprintable material. Thus, instead of "dECM hydrogels as bioinks" I would write "bioinks derived from dECM".
Line 241: cell needs => needs of cells
Line 259: I would delete "with care not to obstruct the hydrogel extrusion channels".
Line 266: What is meant by "an excessive concentration of bioink"? Perhaps the polymer concentration in the hydrogel component of the bioink? Please clarify.
Lines 315-316: Please revise the sentence "Among ... viscosity." Printability is not a mere function of viscosity. Shape stability results from the material's yield stress. See, e.g., [9]
[9] Cooke ME, Rosenzweig DH. The rheology of direct and suspended extrusion bioprinting. APL Bioengineering. 2021;5(1):011502.
Line 568: in vivo constructs => implantable constructs
In my opinion, the English usage and style are fine. Just the specific terminology of 3D bioprinting needs to be revised to assure a consistent communication throughout this rapidly evolving field.
Author Response
Comments:
This manuscript is a narrative review of methods devised to optimize bioinks derived from natural sources. Chemical modifications that seek to reinforce natural-derived bioinks are well described. In contrast, the introductory sections (1 and 2) need a thorough revision to bring the paper in the conceptual framework of the field. The major weak points of this text are the following:
- The language of the manuscript needs refinements to be consistent with the evolving professional jargon of 3D bioprinting and biofabrication. For example, the definition of 3D bioprinting given on lines 33 and 58 deviates from the literature (see refs. [1,2] below). I would encourage the authors to read, the following papers by leading experts (I'm not a coauthor of these works):
[1] Guillemot F, Mironov V, Nakamura M. Bioprinting is coming of age: Report from the International Conference on Bioprinting and Biofabrication in Bordeaux (3B'09). Biofabrication. 2010;2(1):010201.
[2] Groll J, Boland T, Blunk T, Burdick JA, Cho D-W, Dalton PD, et al. Biofabrication: reappraising the definition of an evolving field. Biofabrication. 2016;8(1):013001.
[3] Groll J, Burdick JA, Cho DW, Derby B, Gelinsky M, Heilshorn SC, et al. A definition of bioinks and their distinction from biomaterial inks. Biofabrication. 2018;11(1):013001.
[4] Moroni L, Boland T, Burdick JA, De Maria C, Derby B, Forgacs G, et al. Biofabrication: A Guide to Technology and Terminology. Trends in Biotechnology. 2018;36(4):384-402.
We thank the Reviewer for their comments and suggestions. We edited the introduction section, adding more precise definitions and re-phrasing the concepts properly. All changes are visible in red from line 33 to line 70.
- The perspectives of 3D bioprinting, in my opinion, are overstated by the authors. In particular, the sentence "in a few years will allow the construction of patient-specific organs useful for transplantation purposes" is way more optimistic than the current literature:
[5] Sun W, Starly B, Daly AC, Burdick JA, Groll J, Skeldon G, et al. The bioprinting roadmap. Biofabrication. 2020;12(2):022002.
[6] Ng WL, Chua CK, Shen Y-F. Print Me An Organ! Why We Are Not There Yet. Progress in Polymer Science. 2019;97:101145.
We modified the sentence following the Reviewer’s suggestion:
“All these aspects not only make 3D bioprinting the means that in the future will allow the construction of patient-specific organs useful for transplantation purposes, but it is an in vitro approach that exceeds standard 2D culture techniques [10] and can also eliminate the usual adoption for animal tests, also avoiding the limited accuracy in predicting human toxicological and pathophysiological responses [11]”.
- Section 2, focusing on merely three bioprinting techniques, represents a 10-years-old view. Please revise it to provide a state-of-the-art picture of the field. Good references in this respect are [5] and [7]:
[7] Ji S, Guvendiren M. Complex 3D bioprinting methods. APL Bioengineering. 2021;5(1):011508.
The Authors thank the Reviewer for the suggestion. We modified and expanded Section 2.1, adding more details related to 3D printing in the biological field.
- In sections 3-5, I would cite and discuss the recent review by Prof. Cho's group:
[8] Chae S, Cho D-W. Three-dimensional bioprinting with decellularized extracellular matrix-based bioinks in translational regenerative medicine. MRS Bulletin. 2022.
The suggested article was added as reference number [109] in section 3 (lines 233-235).
“For a detailed overview of 3D bioprinting with dECM bioinks, see the review article of Chae et al. [109] in which the Authors summarized various dECM-based bioink formulations and their tissue engineering applications.”
- To boost the attractivity of this review, please include at least one multi-panel figure in Section 3 to illustrate current applications of dECM-derived bioinks. As it stands, the paper is poorly illustrated. Approval to reproduce original figures from, e.g., refs. 86-93 can be obtained online via RightsLink (click on links such as "Request Permissions" or "Get Rights and Content" next to the abstract of the chosen paper).
We thank the Reviewer for their suggestion. Given that the publication permission of one or more figures published in the identified articles resulted excessive expensive, we followed the reviewer's suggestion by inserting an additional diagram representing the tissues generated through 3D bioprinting using dECM-derived hydrogel as bioink. Figure 1, line 263.
Minor remarks and revisions, listed in the format <original text> => <proposed revision>
We thank the reviewer for their careful and punctual reading of the manuscript. We modified and clarified point by point all the listed suggestions.
Line 35: most ideal => ideal
Line 36: Several are the possible benefits of this organ and tissue manufacturing: => Benefits of this organ and tissue manufacturing include:
Line 52: might result opposite => might differ
Line 136: What do the authors mean by "good resolution"? Please be more specific.
Line 151: secreted and synthesized => secreted
Line 162: Natural-derived bioink advantages: => Advantages of natural-derived bioinks:
Line 216: According to [3], the term "bioink" refers to a cell-laden bioprintable material. Thus, instead of "dECM hydrogels as bioinks" I would write "bioinks derived from dECM".
Line 241: cell needs => needs of cells
Line 259: I would delete "with care not to obstruct the hydrogel extrusion channels".
Line 266: What is meant by "an excessive concentration of bioink"? Perhaps the polymer concentration in the hydrogel component of the bioink? Please clarify.
Lines 315-316: Please revise the sentence "Among ... viscosity." Printability is not a mere function of viscosity. Shape stability results from the material's yield stress. See, e.g., [9]
[9] Cooke ME, Rosenzweig DH. The rheology of direct and suspended extrusion bioprinting. APL Bioengineering. 2021;5(1):011502.
We expanded the section 4 including (lines 288-292):
“The shape stability of the printed construct also closely depends on the yield stress, a critical shear stress value below which a material acts as a solid and above which it flows like a liquid [129]. This is an important feature of bioinks, which will flow as liquid to be printed in a controlled manner. However, as for viscosity, higher yield stress requires increased extrusion pressures, which can negatively impact cell viability.”
Based on this definition, as suggested, we modified line 350 accordingly.
Line 568: in vivo constructs => implantable constructs
Comments on the Quality of English Language
In my opinion, the English usage and style are fine. Just the specific terminology of 3D bioprinting needs to be revised to assure a consistent communication throughout this rapidly evolving field.
We revised the specific terminology of 3D bioprinting, according to the suggestion. All changes are visible throughout the manuscript highlighted in red, and especially from line 33 to line 70.
Round 2
Reviewer 2 Report
The authors have addressed most of the comments in a satisfactory manner, the revised manuscript can be accepted in present form
The quality of English Language is acceptable.
Reviewer 3 Report
This manuscript has been revised properly, taking into account most of my comments. I see no technical reasons to delay the publication of this work. Nevertheless, had it been my paper, I would have tried to assemble a multi-part figure from original articles on applications of 3D bioprinting based on dECM. Some of them appeared in open-access journals, so the permission for reproducing figures is granted by default. Others appeared in journals held by STM Signatory Publishers, which grant permission at no cost for up to 3 illustrations via RightsLink (https://publishingsupport.iopscience.iop.org/stm-permissions-guidelines/). The decision is up to the authors.